# Understanding the Needs and Priorities of People Living with Persistent Pain and Long-Term Musculoskeletal Conditions during the COVID-19 Pandemic—A Public Involvement Project

**DOI:** 10.3390/healthcare10061130

**Published:** 2022-06-17

**Authors:** Sally Fowler Davis, Helen Humphreys, Tom Maden-Wilkinson, Sarah Withers, Anna Lowe, Robert J. Copeland

**Affiliations:** 1Organisation in Health and Care, Advanced Wellbeing Research Centre, Sheffield Hallam University, Sheffield S1 1WB, UK; h.humphreys@shu.ac.uk (H.H.); tm7800@exchange.shu.ac.uk (T.M.-W.); a.lowe@shu.ac.uk (A.L.); slmrjc2@exchange.shu.ac.uk (R.J.C.); 2Sheffield Teaching Hospitals NHS Foundation Trust, Glossop Road, Broomhall, Sheffield S10 2JF, UK; sarah.withers@nhs.net

**Keywords:** experience, public involvement, narrative reporting, underserved communities, chronic pain, COVID-19 pandemic

## Abstract

Background: Critiques of public involvement (PI) are associated with failing to be inclusive of under-represented groups, and this leads to research that fails to include a diversity of perspectives. Aim: The aim of this PI project was to understand the experiences and priorities of people from three seldom-heard groups whose musculoskeletal pain may have been exacerbated or treatment delayed due to COVID-19. Engaging representatives to report diverse experiences was important, given the goal of developing further research into personalised and integrated care and addressing population health concerns about access and self-management for people with musculoskeletal pain. Methods: The project was approved via Sheffield Hallam University Ethics but was exempt from further HRA approval. A literature review was conducted, followed by informal individual and group discussions involving professionals and people with lived experience of (a) fibromyalgia pain, (b) those waiting for elective surgery and (c) experts associated with the care home sector. Findings from the literature review were combined with the insights from the public involvement. Resulting narratives were developed to highlight the challenges associated with persistent pain and informed the creation of consensus statements on the priorities for service improvement and future research. The consensus statements were shared and refined with input from an expert steering group. Results: The narratives describe pain as a uniformly difficult experience to share with professionals; it is described as exhausting, frustrating and socially limiting. Pain leads to exclusion from routine daily activities and often resigns people to feeling and being unwell. In all cases, there are concerns about accessing and improving services and critical issues associated with optimising physical activity, functional wellbeing and managing polypharmacy. Exercise and/or mobilisation are important and commonly used self-management strategies, but opportunity and advice about safe methods are variable. Services should focus on personalised care, including self-management support and medication management, so that people’s views and needs are heard and validated by health professionals. Conclusions: More research is needed to explore the most effective pain management strategies, and public involvement is important to shape the most relevant research questions. Health and care systems evaluation is also needed to address the scale of the population health need. The pandemic appears to have highlighted pre-existing shortcomings in holistic pain management.

## 1. Introduction

Public involvement is used to understand the nature of the problem or situation, particularly when there is limited shared understanding across medical and social care. ‘Involve’ [1] defines public involvement in research as being carried out ‘with’ or ‘by’ members of the public rather than ‘to’, ‘about’ or ‘for’ them. This includes, for example, working with research funders to prioritise research, offering advice as members of a project steering group or, as in this case, undertaking interviews with selected ‘expert’ participants. Public involvement is critically associated with providing a safe and inclusive method for participants to engage in a dialogue and be recognised for their contribution [2] as a means of improving research and practice. The democratic tradition of involvement can be seen as a form of co-production in the research process [3] insofar as through inclusive and considered recruitment, the projects highlight a diversity of lived experience of patients, carers and public. With this study, we sought to recognise the critical difficulties and the needs of people with persistent pain, and to focus on their formative contributions to healthcare service design, delivery and research.

## 2. Background

Critiques of public involvement (PI) report failures to be inclusive of under-represented groups [4,5], and this leads to research that fails to include diverse perspectives [6]. People experience persistent pain for multiple reasons associated with disease and disability; the focus of this project was on musculoskeletal pain that may have been under-treated because of more limited services during the COVID-19 pandemic. There was evidence that age, education, income, health status and social support are associated with changes to physical activity during lockdown [7], and this was a factor in managing pain experiences.

Chronic pain is defined as pain lasting for longer than 3 months [8]. Pain affects an individual’s quality of life and the ability to perform activities of daily living, and causes fatigue. It can be a major factor in a person’s loss of independence. Pain can happen across an individual’s life, from childhood through to the very oldest people in our care home system. The British Pain Society refers to pain as a ‘silent epidemic’, impacting the health of individuals and the wider economy [9].

NICE guidance suggests there are strong social and psychological factors associated with pain in addition to biological mechanisms [10], and there is evidence to show that social and economic determinants affect pain experience as well as treatment outcomes, with the most deprived people being more likely to experience chronic pain [10]. These factors suggest the need for interventions to reflect biopsychosocial needs and offer a personalised and holistic approach to care.

The management of pain has been identified as an important patient priority by the James Lind Alliance [11]. Understanding patients’ needs and concerns and the extent to which services currently meet these is crucial to optimising the support available, as well as strengthening patients’ capacity to self-manage their pain.

The aim of the project was to understand the experiences, needs and priorities of people from three seldom-heard and under-represented groups whose musculoskeletal pain may have been exacerbated or treatment delayed due to the COVID-19. The objectives were to highlight peoples’ challenges, experiences and perceptions about how their health could be improved via novel services and interventions, or methods to engage, enable and empower them to manage.

## 3. Methods

The project was approved by the University Ethics Committee (ER33972489) based on the submission of an in-depth public involvement project to study the effect of the COVID-19 pandemic on the experience of people with persistent pain. The first stage was to consult with a local musculoskeletal service and two existing Patient and public involvement (PPI) groups to identify and select three underserved ‘in pain-populations’. Through discussion with the NHS service and the public involvement group in the university research centre, we were able to select three under-served patient populations who were collectively recognised to have had limited access to services during the pandemic. For example, the effect of waiting times in elective care was deemed to be a critical barrier to accessing surgical interventions. Those with fibromyalgia and very old adult populations in care homes were also commonly recognised to have ‘struggled’ to receive specific support and treatment during the ‘lock-downs’ and subsequent restricted service provision.

The public involvement was characterised by two initial methods. Firstly, we performed a literature-based inquiry, to scope the research evidence associated with personal experience of pain, and service engagement for these population groups. Scoping reviews are used to identify knowledge gaps, set research agendas and identify implications for decision making [12]. Then, a series of participant interviews was undertaken using an informal style of conversation to illicit individual perceptions of context, similar to methods used in public policy consultations [13]. These perspectives are related to experience of pain and pain management and service engagement, sometimes over long periods. The interviews were conducted via informal telephone discussions with individuals and groups, and included both professional stakeholders and people with lived experience of pain. The public involvement allowed researchers to access public representatives from existing networks, and extensive field notes were taken during these discussions to record stakeholders’ views on key aspects of pain experience, pain management services, perceived unmet needs and the impact of COVID-19. These field notes were then collated by two members of the research team, and used to shape narrative accounts of the pain experience.

The participants were identified through service activities, including a clinical effectiveness project in the case of the patients with orthopaedic problems, and via networking and the study team’s extensive contacts with community services. The clinical effectiveness project included a short survey to 50 individuals. In all cases, participants were clearly informed that this was a project to identify critical factors and priorities for improvements in service design and insights into new approaches to research. The approaches were made via personal and professional contacts and were accepted as a method normally used where there is little knowledge or limited access to a more marginal group of stakeholders and where access the target population is needed for the project [14].

The targeted participant groups were as follows:People with fibromyalgia via community and voluntary sector groupsFibromyalgia is a long-term condition, where pain is felt all over the body and is thought to be neural in origin. Fibromyalgia is estimated to affect 1.8–2.9 million people in the U.K. [15]. The condition can be diagnosed after people experience severe pain for longer than 3 months, associated with increased fatigue, poor sleep and cognitive decline. Care home residentsOlder adults living in care homes may have entered care with existing pain, but this can be exacerbated by sedentary lifestyles or by deterioration and terminal illness. Enhancing Health in Care Homes [16] sought to improve holistic care for very old adults living in care homes, but the lockdown reduced access to pharmacological assessment and increased isolation from family and friends. Elective orthopaedic patientsThe number of patients who are waiting for elective surgery has increased dramatically, particularly in orthopaedics. In England, approximately 600,000 people are currently waiting to begin treatment for trauma/orthopaedic interventions [17]. National policy recommends increasing the offer around self-management services [18], but organisations such as National Voice have highlighted other priorities and recommendations from patients currently awaiting treatment, including a desire to be listened to, supported involvement in decision making, clear and accessible information and regular contact with healthcare professionals [19].

### Synthesis by Expert Consensus

Field notes and literature were combined to form ‘narratives’ about the three population groups and combined pain experiences. The narratives seek to form ‘rich descriptions’ based on themes identified from the literature and the reports from the interviews but are not intended to provide a combined view of all pain experience. Population health is based on an approximation of health and the goal of public involvement is to inform and provide a sensitive insight into the needs and aspirations of the group. In the field of population health, divergent opinions are welcomed but need to be combined to enable decision making about how research is conducted and ultimately how care is delivered [20]. To this end, the narratives were also used to generate key themes that cut across the three populations of interest. Two researchers identified and selected specific thematic statements that could then be re-presented and validated by participants, and these were specific points about pain experience that were identified across all the groups.

Finally, all participants, professional stakeholders and people with lived experience who had contributed to the public involvement discussions were invited to a meeting to share the initial findings and provide a greater reach into the ‘pain experience’ through a nominal group method. This approach to group prioritisation was selected because it has advantages over other methods, such as enabling different perspectives to influence the prioritisation of issues [21]. The data synthesis brought people together to identify priorities for improvements in services and in research and to deepen the participation in the analysis, as is the case for complex evaluations [22]. Due to the continuing pandemic and the vulnerability of some of the participant groups, all group interactions were held via video conference and the prioritisation was undertaken using an application allowing individual voting and remote consensus building. No statistical interpretation was made of the responses.

## 4. Results

The summary narrative accounts of the pain experience for each of our groups of interest (fibromyalgia, older adults in care homes and people awaiting elective orthopaedic surgery) are presented below. Additional references to the literature are combined with these accounts where appropriate to add explanatory depth. The narrative accounts are meant to share aspects of experiences as described, whilst recognising that not everyone’s experience was the same and highlight some of these differences.

### 4.1. Understanding the Pain Experience for People with Fibromyalgia

Eight people living with fibromyalgia and four professionals working in musculoskeletal services were included in our PPI discussions. The professionals were a GP, a trustee of a fibromyalgia advocacy charity and musculoskeletal/pain clinic specialists. 

Fibromyalgia is a long-term condition, where pain is felt all over the body and is thought to be neural in origin, alongside disordered sensory processing [23]. Several pathogenetic reasons are accepted [24,25,26], including ‘life changes’, particularly gynaecological [27,28,29], trauma or other forms of stress and vulnerability [30]. The most common experience of fibromyalgia is of persistent musculoskeletal pain with increased fatigue, poor sleep, gastrointestinal symptoms and cognitive decline [26,27,28,29,30]. An estimated 1.8–2.9 million people are affected in the U.K. [15].

#### 4.1.1. Living with Pain

People living with fibromyalgia describe their experiences of pain as ‘‘exhausting’’ and ‘‘all-consuming’’; the illness and its associated symptoms affect all aspects of their lives. People feel excluded or cut off from normal life because of unpredictable fluctuations in their health. Levels of pain make it challenging to maintain regular employment and social relationships. This leaves people feeling isolated at home and makes the days hard to fill; hobbies can be difficult to pursue because of symptoms including joint pain and ‘brain fog’.

#### 4.1.2. Communicating Pain

Pain is challenging to communicate, and other people cannot understand how it feels. Pain is both invisible and subjective and it is perceived that other people often underestimate the severity and impact of pain. Friends, colleagues and extended family often only see people with fibromyalgia when they are well and therefore appear not to believe them when they try to express how unwell they sometimes feel. The condition is very individualised and there is a wide spectrum in terms of severity of debilitation.

Other people with fibromyalgia can be a great source of support and validation. Peer support groups and charities play a key role in supporting wellbeing and fill some gaps left by the healthcare system. It can be upsetting to hear about others who are able to engage in daily activities—for example, managing to work full-time. Newly diagnosed people can be confronted with seemingly contradictory advice as well as stories of trauma which are not always constructive. To reinforce this, healthcare professionals have been observed to characterise people with fibromyalgia in having a profile of complaining [31] or describing people as wanting to be ill [32].

#### 4.1.3. Diagnosis

There is a paradox around diagnosis. Diagnosis only happens by excluding other possible causes, and in some cases, this can take several years. Obtaining a diagnosis is dependent on the willingness of health professionals to investigate symptoms as well as individuals’ inclination to pursue multiple, often frustrating, and fruitless conversations, tests and treatments. Many people give up during this process and fall out of the healthcare system, meaning that they are left without answers and miss out on opportunities for support. Those people who eventually do receive a diagnosis may feel some temporary relief, but diagnosis also brings with it a realisation about the many assumptions and stigmas related to fibromyalgia. This includes contentions about whether fibromyalgia is ‘real’, which many people say they are regularly confronted with. Some people feel that the ‘label’ of fibromyalgia has prevented them from being taken seriously when complaining of being in pain. They feel that medical professionals have labelled them as hypochondriacs or assume that they are mentally ill, resulting in a failure to have other health problems (unrelated to fibromyalgia) properly diagnosed.

#### 4.1.4. Lack of Control

Pain can be episodic and unpredictable. When pain flares up, it can take weeks to subside. Consequently, people with fibromyalgia described living moment-to-moment out of necessity rather than choice. Most people said it was impossible to plan and this made it hard to commit to anything. Knowing that there is no treatment or cure and no end to the pain feels exhausting, depressing and at times, some people feel suicidal. They want answers and explanations and are anxiously seeking ways to alleviate their pain. This can include medical, non-medical and ‘alternative’ methods. There are no specific evidenced treatments that have been shown to be effective for all. Some people follow current research outputs, but this can provide false hopes and over time can be disheartening. Those who have suffered a long time with fibromyalgia are stoic and resigned to their adjusted lifestyle—they feel they have no choice but to ‘crack on’. This is not an easy position; instead, it is one of reluctant acceptance.

#### 4.1.5. Managing Activity

Living with fibromyalgia means that people must prioritise every task and carefully manage their energy expenditure. Employment often takes priority because of financial needs, meaning that leisure or personal activities are often sacrificed. Musculoskeletal pain limits options when it comes to the types of work people can do. Pacing helps, but it is not enough to enable people to live ‘normally’. Complete inactivity and too much activity are both problematic, but maintaining the ‘right’ degree of activity can be a very difficult balance to find (activity can make you feel worse before you feel better and therefore tends to be avoided). Many people with fibromyalgia resent hearing that they need to ‘experiment’ to find the right level of activity because doing so can be exhausting and demoralising. Exercise is also a sensitive subject because of negative experiences people have had with ‘graded exercise’. Physical activity-based pain management strategies should be considered based on patient preference [30] as well as integrative conversations around motivations and barriers to exercise, and some activities such as yoga [33] and tai-chi [34,35] have been shown to be effective for pain management.

#### 4.1.6. Access to Services

People with fibromyalgia do not feel that they fit anywhere in the current healthcare system; appropriate, specialist pathways do not exist. Fibromyalgia pain is complex and multifaceted, yet the primary care system is only set up to address symptoms in isolation. One person described having to play ‘top trumps’ with symptoms when visiting the doctor, selecting the most problematic or the ‘most interesting’ symptom to share. Time-limited medical appointments are insufficient for professionals to provide a person-centred, holistic response. The trial-and-error nature of the medical approach leaves people feeling exasperated and unheard; many have been referred back and forth between primary care and rheumatology. They either disengage altogether or develop a poor opinion of medics. This sours the future therapeutic relationships and makes them feel mistrustful of any support that is later offered. There seems to be a postcode lottery at play: geographical variation in local service availability, e.g., social prescribing, community organisations.

The use of pain clinics reduces clinical needs of patients [36], and from a patient perspective, there is a higher acceptability of non-pharmacological interventions [37], despite similar levels of effectiveness with pharmaceutical interventions [38]. Ashe and colleagues [35] conducted a qualitative study into patient experiences of living with and treatment experiences of fibromyalgia, with patients interviewed liking the interdisciplinary nature of pain clinics and helping them to ‘engage in a nurturing therapeutic relationship...creating a fibro-family.’ This provides an example of the impact of an ‘expert patient’ where individual self-management of pain might be more effective, especially when supported by peers [28,35,39,40,41].

#### 4.1.7. Primary Care

People with fibromyalgia can experience a lack of understanding and acceptance from the medical community. GPs can vary in their attitudes towards [42] and knowledge about fibromyalgia and it is common for people with fibromyalgia to move GP surgeries to find someone more supportive. It is difficult for people with pain to travel because mobility can be very limited, but existing services are often beyond the comfortable range of travel. This lack of understanding from medical professionals and the general public may make symptoms worse [39] and has left people with fibromyalgia feeling stigmatised [43]. There is a tendency towards pharmacological treatment and variable access to holistic pain support [37].

Some people with fibromyalgia have concerns about the use of pain medications, particularly side effects and possible addiction, and some professionals share these concerns. Avoidance of and/or over-reliance on pain medication is a concern for some patients, along with differing perspectives towards physical activity and other self-management strategies [44]. Out of desperation for pain relief, some people have spent vast amounts of their (already limited) income with little guidance or support from medical professionals. Many have benefited from some form of psychological or cognitive therapy. However, they often need multiple opportunities or offers to engage with therapy so that they can benefit when they are ready for it.

#### 4.1.8. Impact of COVID-19

Initially, COVID-19 provided some security and relief through shielding; people with fibromyalgia welcomed that everyone was in the same boat. Unfortunately, the consequences of longer waiting lists and limited face-to-face contact with healthcare professionals means that they are feeling more uncertain and isolated. Going out now makes people feel anxious because they worry about risk of COVID-19 transmission. Changes to benefits assessments following COVID-19, e.g., eligibility for higher rate personal independence payments, have had a particularly negative impact. Some people feel that they effectively have to behave like a ‘one-person campaign’, advocating for themselves within both the benefits and healthcare systems, which is draining and demoralising.

#### 4.1.9. Differences in Experiences by Demographic

People who do not speak English well may be disproportionately affected by the issues presented above, as they may lack the health literacy and/or language skills to advocate and self-manage. Professionals report that medical terminology about pain does not translate across well to different languages or cultures. This makes it very difficult to access relatable information and support, and/or to digest advice about self-management.

It seems that males tend to under-report pain and symptoms of fibromyalgia and might also be less likely to pursue a diagnosis. This means that many men might not be accessing support. On the other hand, many females with fibromyalgia feel that they are not taken seriously. They believe that their symptoms are unfairly blamed on hormones or poor mental health. Professionals know that links exist to these aspects of health, but they are not well understood. Fibromyalgia diagnosis is nearly 4 times more prevalent in females than males [29], with some reticence in medical professionals to diagnose males with fibromyalgia reported, much to the frustration of patients [42].

### 4.2. Understanding Pain Experienced by Elective Orthopaedic Patients

Seven people awaiting elective orthopaedic surgery agreed to discuss their pain experiences, plus two professionals (a nurse and an occupational therapist).

The number of patients who are waiting for elective surgery increased dramatically during the pandemic, and the longer patients live with pain, there is a risk of reduced functioning and decreased mobility through muscle wasting [45]. Waiting times are the biggest cause of dissatisfaction within the NHS [46,47]. Long waiting lists for elective surgery defined as greater than the targeted 18 weeks for treatment [48] have been associated with worse health perceptions [49], deterioration in quality of life and raised anxiety [46,47], as well prolonging sick leave [50,51,52,53,54] and loss of income [49,50,51,52]. International agencies (The European Hip Society and Knee Associates) and national groups (Versus Arthritis) have emphasised the need to assess the impact of delaying hip and knee arthroplasty and prioritise waiting lists [53,54,55,56].

#### 4.2.1. Beliefs about Elective Surgery and Pain

People we spoke to were accepting of and resigned to the long wait for surgery. This might be because they are used to seeing media reports about long waiting times or know someone else who has waited several years for surgery and/or because they understand that waiting times are significantly affected by COVID-19. Despite general fortitude about waiting times, people would find it useful to have more information about how long they might expect to wait, or a general idea of their position in the ‘queue’. Most people waiting for elective surgery also feel resigned to having pain. This is because pain is often normalised as an inevitable aspect of waiting for surgery and they therefore regard it as something to be tolerated.

#### 4.2.2. Impact of Pain

Despite stoicism and pain tolerance, pain can have a major impact on peoples’ daily lives. For several people, their pain causes mobility problems which have resulted in them having to stop work. Depending on the type of employment they had, this varies from being signed off on long-term sickness absence to taking voluntary redundancy. Whilst pain is the main reason for stopping work, depression might also be a contributing factor for some. It is not easy to tell whether this depression is directly related to the pain or is something separate.

For most people, pain is variable and can range in severity on different days. Whilst this is largely tolerable, it does mean that they need sufficient knowledge about how to effectively manage their pain on any given day. For some people, pain causes disruption to sleep, which also affects quality of life.

#### 4.2.3. Communication with Healthcare Professionals

People describe difficulties accessing their GP as well as physiotherapists and other professionals. Some would have liked to see a health professional for a face-to-face appointment, and they have struggled with the lack of access to professionals during the pandemic. Seeing professionals on a one-to-one basis is reassuring, and generic mailings or information are no substitute for this. In addition, receiving a ‘keeping-in-touch’ information mailing is very welcome, mostly because people appreciate ‘feeling remembered’ by the health service. Patients welcome access to more information about their situation, including what to expect and when.

#### 4.2.4. Self-Management of Pain

Most people already have pain management strategies in place, which for some, tends towards medication. Those who have had pain for a long time feel that they have been ‘through it all before’ with their GP and have exhausted any possible options for reducing their pain. Pain medication has side effects which can be significant. Information is power—letting people know early in their wait what they can do to manage their pain would be best. Most people are very willing to be proactive in managing their pain and welcome any information or advice that is offered. More information would be particularly welcome about non-pharmacological self-management strategies, including exercise and the use of heat and ice. People want to know not just what to do but why it is going to be useful. Generic information is acceptable (i.e., from the internet), but many would prefer that it is personalised, to give confidence that it is ‘right for me’.

Exercise and/or movement are among the most used self-management strategies for pain. Some patients take the initiative and experiment to work out which physical activities work for them. Others would welcome more information about what type of exercise is right and safe to do, and some would like to be supervised while they exercise. Regardless of their level of confidence, everyone would welcome more reassurance and guidance about how to use exercise and activity in the management of their pain.

#### 4.2.5. Differences in Experiences by Demographic

The patient’s perspective of being on long waiting lists for elective surgery has received relatively little attention [57], but the National Voices’ [19] whitepaper on improving our understanding of the experience of waiting for elective care included an extensive literature review to understand factors that impact patients’ experiences. Individuals’ experiences become worse over time, with pain affecting sleep or activities of daily living, particularly with older patients. The British Orthopaedic Association reported that between April and September 2020, there were 212,000 (73%) fewer operations during the first surge of the pandemic, but also that many people avoided healthcare settings due to perceived risk of infection of COVID-19 [58].

### 4.3. Understanding Pain of Older Residents in Care Homes

Six individual discussions took place with professionals working with older adults in care homes. This included a deputy care home manager, a care home pharmacist and a GP with specialist interest in care homes. These professionals shared rich insights into the management of pain in care homes but were not able to report lived experiences. During the pandemic, it has become more difficult to engage people working in care homes or residents and their family carers.

Older residents have difficulties expressing and communicating pain, and self-reporting pain with residential care populations is a key challenge for the sector. Residents are dependent on the professionals and family/friends around them to understand their experience by virtue of their behaviours, and there are often subtle signs of pain (e.g., body language, facial expressions, sounds, agitation, food and fluid intake, signs of movement distress and restlessness at night) that make it evident that they are not comfortable. Reports of care staff feeling ‘traumatised’ and exhausted since the pandemic and lockdown resulted in an overall drop in activity but also exacerbated the previously endemic problem of ‘normalised acceptance’ of pain. Residents have suffered through prolonged periods of isolation and heightened confusion caused by disruption to normal routines and carers wearing personal protective equipment, but some are experiencing ‘unbearable pain’ that they cannot express, or which is not fully managed within their care plan.

The prevalence of pain in older people in the community is 25–76% [59,60,61,62], whereas in residential care, it is assumed to be far greater, at 62–93% of the population [63,64,65,66]. Pain is a major burden of illness experienced by older people in care homes, but it is unclear whether pain is based on disease processes or resulting from the sedentary lifestyle experienced within these institutions [67]. Older adults in care homes are prone to physical inactivity and sedentary behaviour [68], and older adults and the care workforce often collude in promoting inactivity [68] with, for example, ‘wandering’ activity perceived as unsafe and disruptive [69]. There have been several large-scale trials associated with changing practices to improve physical activity in care homes, with mixed success, more often reporting some of the barriers to changing practices that are intended to improve quality of life [70,71].

#### 4.3.1. Assessing Pain Experience

Some professionals suggest that it is common for older residents to understate or diminish their pain. Stoicism and/or beliefs that pain is a normal and inevitable consequence of ageing appears to result in many older residents not mentioning it unless explicitly prompted. Older people frequently dismiss musculoskeletal pain as ‘just a niggle’ or nothing more than what they ‘usually’ experience. When staff hear about pain very frequently, they can become desensitised to it and/or the issue can become normalised for that patient. This makes it challenging for staff to assess, and for the patient to communicate, when changes in their pain occur (e.g., escalation of existing pain or new pain).

Professionals highlighted the importance of understanding the emotional component of pain. Identifying triggers and sources of pain and distress is key—but not always simple. Professionals such as doctors, nurses, pharmacist, dentists and physiotherapists who are more likely to address the pain are heavily reliant on the observations of the carers, which can be a subjective assessment of the pain by the observer. Significant time can be necessary to investigate and understand an individual’s pain, but many care professionals and visiting health professionals lack adequate time to undertake in-depth assessment or consultations. Interactions with external medical professionals (GP, district nurses) are particularly brief and often (currently) virtual, making it even more challenging to assess and identify pain-related issues [72,73,74].

Older adults can hold attitudes, beliefs and expectations affecting pain reporting [59], with stoicism particularly prevalent in the care home population. This stoicism is linked to under-reporting of pain [75], with numerous studies evidencing age-related differences in stoicism, pain reporting and chronic pain experience [76,77,78,79]. With dementia, the neuropathological changes disrupt neural pain networks [80], especially those involved in pain perception, and there is some evidence that people with dementia fail to recognise and evaluate their own pain [81,82], making treatment all the more complex [83].

#### 4.3.2. Communicating Pain Experience

The ability to communicate distress depends on the degree of cognitive, sensory and/or speech impairment an older person experiences; for example, people with dementia can sometimes display aggression or agitation, which may be because pain has worsened and is not taken seriously or sufficiently investigated. Recent work to understand facial expression of pain [84,85] using the facial action coding system [86] rather than self-reported outcome measures has been successful, with facial responses in dementia patients [87] leading to non-verbal cues [88,89].

Individual differences in pain tolerance, beliefs about pain and preferences for its management are not obviously reflected in current management strategies with an apparent emphasis on pharmaceutical management. Residents often consider their role in pain self-management to be medication adherence but with little knowledge or critical reflection on what they are taking and why. Patients have limited, if any, access to other supportive therapies, including exercise-based rehabilitation, physiotherapy and occupational therapy. Therapy is generally limited to the provision of equipment for safety purposes.

#### 4.3.3. Lack of Agency and Access to Services

Older care home residents have very little agency when it comes to their living situation, and this extends to how their pain is managed. Older people spend large amounts of time sedentary and/or in bed with muscle wastage, nerve compression, postural issues and stiffness, all potentially contributing to increased pain. Mobilisation is an important aspect of pain management, but older people are not always able to initiate movement themselves. Care home protocols prioritise safety and the minimisation of fall risks, meaning that residents are moved infrequently and those who can stand, transfer or walk with support are sometimes discouraged from doing so (e.g., by hoisting). Professionals indicated high levels of depression in care home residents, partly related to boredom, neglect and/or lack of stimulation. Limited (nil, in many cases) engagement from family members can be one of the contributory factors to depression and psychosomatic pain for many older people, highlighting the social aspect of care.

Long service waiting times on referrals can have profound adverse effects for care in terms of delayed identification of the issues. Heat relief, cold compresses, distraction therapies and mobilising or at least increased repositioning can all be useful non-pharmaceutical strategies, but are not used widely enough.

#### 4.3.4. Complexity of Pain Medication for Older Residents

Physical and mental health co-morbidities for older people often manifest into polypharmacy [90]. Inappropriate or high-dose pain relief medication(s) adds to the polypharmacy, which can potentially aggravate kidney damage, oversedation and confusion, constipation, rebound pain and increase risks of falls and fractures. Oversedation or withdrawal from pain medication can exhibit symptoms which can be misinterpreted (e.g., as psychiatric symptoms, with possibly more medication being prescribed), and people in residential care are more susceptible to side effects from polypharmacy, especially delirium [91]. Professionals described a ‘sundown syndrome’ for residents with dementia in which pain becomes worse in the evening and nighttime, causing increased aggression, restlessness and sleep disturbances and cognitive impairment [73], potentially resulting in undermedication [92,93]. This can lead to hypnotic prescribing, whereby hypnotics are prescribed alongside strong opiates. Such medication can cause psychiatric symptoms and/or a hangover effect the next morning, with increased risk of falls.

Older people discharged into care straight from acute hospital settings can tend to be prescribed strong pain relief. This is not intended to be a permanent dosage, but intentions to reduce this once under primary care management are often thwarted by a lack of GP time to conduct proper follow-ups and medication reviews, inadequate pain care planning resident or family misperceptions about pain medication (i.e., stronger = better). This results in residents remaining on strong pain relief with significant side effects when pain could be reasonably managed through less intrusive medications and other strategies. A lack of understanding and demands for medication by the carers and the family often put pressure on the professional, making it difficult to reduce the inappropriate prescribing.

#### 4.3.5. Lack of Pain Care Planning and Documentation

Pain management in care homes is largely reactive, but it could be more effective if tackled pre-emptively.

Older people in care homes, particularly with memory loss or dementia, need regular objective pain assessments and documentation to aid holistic medication review and care. It is difficult for professionals to monitor what is working/not working in terms of pain management because it is usually poorly documented. Professionals suggest that proper pain management care plans with specific documentation for adoption in care homes, clear recommendations and escalation plans are warranted. Digital care plans and/or quick access to the correct documents may further help professionals. Better training and consistent use of pain assessment tools may also be warranted.

#### 4.3.6. Differences by Demographic

There are clear sociodemographic influences on pain, with examples of military veterans who are considered to have a higher pain tolerance compared to the general population [94]. There also appears to be some influence of spiritual or religious beliefs having positive outcomes for pain [95,96]. Studies show that women have a higher prevalence of pain [59,97].

#### 4.3.7. Effects of COVID-19

There have been enhanced measures to protect and isolate the very old and vulnerable from infection [98], but inevitably, the specific needs of individuals have been neglected in relation to under assessment and reduced access to specialist care during the COVID-19 pandemic [84].

### 4.4. Findings from the Synthesis of Pain Experiences

Two members of the team collated and identified cross-cutting statements from the three public involvement narratives, grouped as (i) statements about pain experiences, (ii) service improvement priorities and (iii) future research priorities. The three lists of statements are included in Appendix A.

The nominal group method [99] was conducted online with participants using an online polling tool that enabled individuals to prioritise the statements. Twelve participants attended, including professional experts and people with lived experience from across our three identified subgroups (fibromyalgia, care home residents and waiting elective orthopaedic patients). The statements were discussed and some minor refinements were made to the language based on feedback from participants. Statements were then sorted and combined into a list of priority areas.

The top five final statements, prioritised by participants, are as follows.

#### 4.4.1. Pain Experience

Pain experience is difficult to communicate to professionals and others.Pain is exhausting, frustrating and socially limiting.Pain results in exclusion from usual daily activities, e.g., regular employment.Pain makes you resigned to being/feeling unwell and may be constant and enduring.Pain is poorly understood by some health professionals, and this affects trust.

#### 4.4.2. Service Statements

Everyone with pain should have an individually tailored care plan that includes supported self-management and pain medications.People with pain most want to feel heard and validated by health professionals.Pain is different in intensity for everyone, so it needs to be properly and continuously assessed.It is necessary to consider lots of different approaches that alleviate or make living with pain easier.Pain should include identifying and addressing emotional needs, triggers and sources of pain.

#### 4.4.3. Research Statements

More research is needed to inform professionals’ prioritisation of effective pain management strategies that tailor the care plan to the individual.Professionals and agencies need guidance on age-appropriate and culturally appropriate opportunities for people to learn to manage pain.Healthcare professionals need evidence-based knowledge and training about the perception of pain medicine.Best practices in pain assessment tools and implementation of continuous assessment are required.Professionals and agencies need guidance on how to communicate about pain.

## 5. Discussion

This public involvement has confirmed that people with chronic pain are often, in different ways, unable to access the services they need, and as such, their needs are ‘seldom heard’ and therefore underrepresented in commissioning decisions and service planning. By creating narratives developed from this public involvement project involving and critically identifying the needs of three underserved groups, the aim was to describe concerns and prioritise needs for care, support and further research. People living with chronic pain have difficulty describing their experience and the functional limitations that affect their quality of life. It is almost always associated with depression and significant limitations to social and work activity.

For people with fibromyalgia, there is a highly varied understanding of what works and for whom, and a ‘reluctant acceptance’ that very careful and supported self-management is needed. For older adults in care settings, pain is ‘accepted’ and undertreated. In people experiencing long waits for musculoskeletal surgery, there is a high level of stoicism, but many would benefit from advice about how to maintain functional wellbeing. In all cases, there are concerns about accessing and improving services and critical issues associated with optimising support for physical activity and functional wellbeing and about managing polypharmacy.

The COVID-19 pandemic has exacerbated perceptions of isolation for all these population groups and increased barriers to accessing support, particularly through reduced access to face-to-face contact with health professionals and increased demand for psychological therapy and welfare benefits. People with long-term conditions report more psychological symptoms than the general population [100], with social isolation further exacerbating symptoms and potentially adversely impacted pain-related treatment outcomes [101].

Older people and people with chronic disease, those from lower income groups and those concerned with acquiring COVID-19 develop stress, anxiety and depression, leading to a higher risk of persistent pain [102,103,104,105]. Indirect reports of the chronic pain experienced by older frail people (in particular) included ideas associated with learning to recognise a non-verbal pain response and a more in-depth assessment of needs associated with managing pain through medicines optimisation and reducing the very sedentary lifestyle of many residents.

These narratives, with additional literature, support the work of ‘National Voices’, who have recommended particular improvements for future practice [19]

Better communication in managing patients’ expectations.

Accurate information for whether to wait for NHS services or pay for private services.Providing status updates to ease anxiety, especially in patients with low social support. Gillis et al. [106] reported that patients hold off making large financial commitments or bookings for things such as holidays or family visits until they have their surgery.Letters are out of date and more up-to-date two-way communication is needed.Information on what is being done to manage waiting lists and reduce their waiting times.

In addition, there is a reported 35% difference in waiting times between the most and least deprived population quintiles [107]. This is attributed to a number of reasons, including (a) elbowing behaviour—individuals who are least deprived are more likely to have access to information, better networking skills and contacts enabling them to apply more effective pressure to secure priority for treatment [106]; (b) defensive medicine—avoiding the threat of legal action; and (c) unconscious bias, based on understanding the needs of those in similar socioeconomic status to themselves. [107]

An important finding from this study is a perception that patients and healthcare professional can hold maladaptive beliefs that increased pain with a certain activity is harmful, and this can result in activity avoidance. The perception that pain is unmanageable can further increase the effects of deconditioning and further exacerbate pain [108,109]. Fear of pain and avoidance behaviours are common with painful MSK conditions [86,87], and a more careful, person-centred approach to physical activity is warranted to enable patients to confidently remain active and alleviate musculoskeletal pain.

## 6. Limitations

Our public involvement activity highlights clear challenges in communicating chronic and disabling pain, but there is a lack of published literature about the experiences of pain from people in seldom-heard groups. Further targeted research that includes and involves practitioners and patients would be needed to scale up the findings here, so that greater confidence in the population view could be assured. Further diverse voices are also needed so that tailored interventions can be co-produced with these communities. This project was an opportunity during the COVID-19 pandemic to scope the needs of people with persistent pain experience and as such did not include interventions or measures to report. The main limitation was in the access to residents and direct experience of older people with pain knowledge. Further systematic research is needed to identify the service barriers to managing chronic MSK pain across this sector.

## 7. Conclusions and Recommendations

This project used public involvement with professional and expert stakeholders with ‘lived experience’ to prioritise issues related to pain and pain management in under-served populations during the COVID-19 pandemic. In addition, an embedded literature review aimed to amplify the lived experience of chronic pain across three underserved populations during the COVID-19 pandemic. The findings, presented as three narrative accounts, highlight the difficulties of explaining pain to others and engaging fully with services. These insights should be used to inform research questions, orientating researchers to clinical and functional issues for those living with pain. Using a nominal group method, the final statements suggest that research (and researchers) must demonstrate compassion for the enduring pain experienced by sufferers and undertake studies that specifically address the needs of those who have culturally diverse or age-appropriate needs, and that research into the competencies of those who treat people in pain is needed.

This public involvement project achieved its aims firstly by comprehensively exploring and amplifying the voices of selected seldom-heard groups, and secondly by conducting a robust process that co-developed a concise summary of potential priorities for future research.

## Data Availability

The datasets generated during the current study are not publicly available due to participants not consenting to this but are available from the corresponding author on reasonable request.

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
