# Peer review of "Understanding the Needs and Priorities of People Living with Persistent Pain and Long-Term Musculoskeletal Conditions during the COVID-19 Pandemic—A Public Involvement Project"

_healthcare, 2022, doi:10.3390/healthcare10061130_

Round 1

Reviewer 1 Report

Congratulations to the authors for this manuscript, the intended purpose was to understand the experiences and priorities of people from three seldom heard groups whose musculoskeletal pain may have been exacerbated or treatment delayed due to COVID-19.

My comments and suggestions for changes to the article are the following:

1º Indicate the type of study performed in your title and abstract.

2º Enter keyword: COVID-19 Pandemic or COVID-19.

3º The Plain Language Summary and Background could be replaced by an introduction that explains the following elements: theoretical and empirical background on the research topic; the research question; the formulation of the objectives; the main variables or dimensions of the object; the hypotheses of the study; issues of public importance; previous qualitative research that has been conducted on the same topic, both in the same cultural and social context and in other different contexts; it may be enriching to contextualize the social relevance of the topic addressed.

4º Attach the COREQ checklist for qualitative studies and you will find that there are some sections within your methodology that are incomplete such as:

4.1º What methodological orientation was established to support the study? For example, grounded theory, discourse analysis, ethnography, phenomenology, content analysis.

4.2º What are the important characteristics of the sample? demographics, date, should be indicated.

5º There is a lack of information regarding data collection:  Did the authors provide question guides, prompts? Was the interview/focus group protocol piloted? Were repeated interviews conducted with the same participant? If so, how many? Was audio or visual recording used to collect data? Were field notes collected during and/or after the interview/focus group? What was the duration of the interviews/focus group? Was data saturation discussed? Were transcripts returned to participants for comment and/or correction? The questions I raise need to be specified within the methodology used in conducting the interviews.

6º You present a very interesting qualitative study with a bibliographic review that reinforces your data collected from the selected populations. But like any qualitative study, the following sections should appear in the methods section in a clear and defined manner: study design; participants and context; data collection; data analysis; ethical aspects and rigor.

7º Indicate all the information related to the ethics committee, the ethics committee approval number does not appear, this information should appear in the methods section.

8º Discussions should encompass the key findings of the study: discuss any previous research related to the topic to place the novelty of the discovery in the appropriate context, discuss possible shortcomings and limitations in its interpretations, discuss its integration into the current understanding of the problem and how. This advances current views, speculates on the future direction of the research, and freely postulates theories that might be tested in the future, completed and reformulated. You make a good analysis of your results, but there is a lack of information in reference to previous research related to the subject, you could broaden the discussion in this very determinant aspect.

9º You employed a nominal group technique to facilitate idea generation and problem analysis, in your findings you indicate that this technique resulted in specific priorities for future research (line 624-625), could you define in your findings what specific priorities are derived from your analysis?

10º Include in the conclusions a prospective for future researchers in reference to the limitations found or possible deficiencies of the research proposed.

11º Adapt the bibliographic references to the Healthcare journal guidelines.

Author Response

1º Indicate the type of study performed in your title and abstract.-  This was a public involvement project (not primary research) undertaken with key stakeholders to understand and to prioritise research questions - this has been reinforced in the title and abstract

2º Enter keyword: COVID-19 Pandemic or COVID-19.  Done

3º The Plain Language Summary and Background could be replaced by an introduction that explains the following elements: theoretical and empirical background on the research topic; the research question; the formulation of the objectives; the main variables or dimensions of the object; the hypotheses of the study; issues of public importance; previous qualitative research that has been conducted on the same topic, both in the same cultural and social context and in other different contexts; it may be enriching to contextualize the social relevance of the topic addressed.-  This was a study/ project  undertaken to understand the experience of a marginalised population.  As an important pre-cursor to primary research it involves in depth co-production of priorities for research, hence there is no hypothesis and the research question is not needed.  However the plain English section has been removed and the public importance of the topic re-written

4º Attach the COREQ checklist for qualitative studies and you will find that there are some sections within your methodology that are incomplete such as: (please see comments below)

4.1º What methodological orientation was established to support the study? For example, grounded theory, discourse analysis, ethnography, phenomenology, content analysis.  The public involvement was not undertaken with a methodological orientation.  Involvement and engagement have an important but separate rationale.  Participants engage in co-production as equal partners in the development of the priority areas for research to inform and reinforce the views of those with lived experience

4.2º What are the important characteristics of the sample? demographics, date, should be indicated.  This is not needed for PI activity

5º There is a lack of information regarding data collection:  Did the authors provide question guides, prompts? Was the interview/focus group protocol piloted? Were repeated interviews conducted with the same participant? If so, how many? Was audio or visual recording used to collect data? Were field notes collected during and/or after the interview/focus group? What was the duration of the interviews/focus group? Was data saturation discussed? Were transcripts returned to participants for comment and/or correction? The questions I raise need to be specified within the methodology used in conducting the interviews.  This is only relevant if this was qualitative research and not interviews undertaken for co-production

6º You present a very interesting qualitative study with a bibliographic review that reinforces your data collected from the selected populations. But like any qualitative study, the following sections should appear in the methods section in a clear and defined manner: study design; participants and context; data collection; data analysis; ethical aspects and rigor.   The literature is used to substantiate and explain the views of participants and this is not presented as a qualitative study, apologies for any confusion that we have now clarified in the text

7º Indicate all the information related to the ethics committee, the ethics committee approval number does not appear, this information should appear in the methods section.  This has been added 

8º Discussions should encompass the key findings of the study: discuss any previous research related to the topic to place the novelty of the discovery in the appropriate context, discuss possible shortcomings and limitations in its interpretations, discuss its integration into the current understanding of the problem and how. This advances current views, speculates on the future direction of the research, and freely postulates theories that might be tested in the future, completed and reformulated. You make a good analysis of your results, but there is a lack of information in reference to previous research related to the subject, you could broaden the discussion in this very determinant aspect.  Thanks for the comment, our aim was to co-produce a set of critical areas for further research.  It was important to present the voice of the public participants authentically as a position statement in its own right

9º You employed a nominal group technique to facilitate idea generation and problem analysis, in your findings you indicate that this technique resulted in specific priorities for future research (line 624-625), could you define in your findings what specific priorities are derived from your analysis?  This has been clarified and improved in the findings and discussion-  the priorities are identified line 593 onwards

10º Include in the conclusions a prospective for future researchers in reference to the limitations found or possible deficiencies of the research proposed.  Thanks this was an omission and has been added

11º Adapt the bibliographic references to the Healthcare journal guidelines.  Now corrected 

Reviewer 2 Report

This is an interesting study. I would like to recommend its publication. I have several suggestions to make:

First, the type of this study should be clearly defined, for example, a qualitative or quantitative analytic study. 

Second, in Line 139, I believe a number should be given ahead of  "Synthesis by expert consensus".

Third, did the study perform any statistical analysis of their data? Please clarify. 

Fourth, the appendix is suggested to be attached as a part of the supplementary material.  

Author Response

This is an interesting study. I would like to recommend its publication. I have several suggestions to make:

Many thanks for the supportive review and we have addressed concerns below

First, the type of this study should be clearly defined, for example, a qualitative or quantitative analytic study. This is a public involvement project and this has been added to the title and throughout the text

Second, in Line 139, I believe a number should be given ahead of  "Synthesis by expert consensus".

Yes, this has been added

Third, did the study perform any statistical analysis of their data? Please clarify. 

We have clarified the consensus method and this did not include statistical analysis

Fourth, the appendix is suggested to be attached as a part of the supplementary material.  

This has been clarified and is appendix not supplementary

Round 2

Reviewer 1 Report

Thank you for your feedback and for taking into consideration the suggested recommendations.